# Quantification of Impact of Land Use Systems on Runoff and Soil Loss from Ravine Ecosystem of Western India

Gopal Lal Meena [1,*], Bira Kishore Sethy [2], Hem Raj Meena [1], Shakir Ali [1], Ashok Kumar [1], Rajive Kumar Singh [1], Raghuvir Singh Meena [3], Ram Bhawan Meena [4], Gulshan Kumar Sharma [1], Bansi Lal Mina [5] and Kuldeep Kumar [1]

1   ICAR—Indian Institute of Soil & Water Conservation, Research Centre, Kota 324002, Rajasthan, India
2   ICAR—Research Complex for NEH Region, Umiam 793103, Meghalaya, India
3   Agricultural Research Station (SKRAU, Bikaner), Sriganganagar 335001, Rajasthan, India
4   ICAR—Indian Institute of Soil & Water Conservation, Research Centre, Agra 282006, Uttar Pradesh, India
5   ICAR—National Bureau of Soil Survey and Land Use Planning, Regional Station, Udaipur 313001, Rajasthan, India
*   Correspondence: gopal.meena1@icar.gov.in; Tel.: +91-9413800561 or +91-7734928006

**Abstract:** Ravine and gully formations are both spectacular and also the worst forms of water-induced soil erosion and have in situ and ex situ impact on geomorphology, hydrology, productivity and environmental security, and they are the root causes of degradation of marginal and adjacent land along with reduced production potential. A long-term (2011–2019) study was conducted on marginal land of the Chambal ravine to assess the impact of six land uses, i.e., Agriculture ($T_1$—Rainfed Soybean), Agri-horticulture ($T_2$—Soybean + *Manilkara achras*), Horti-Pastoral ($T_3$—*Emblica officinalis* + *Cenchurus ciliaris*), Pasture ($T_4$—*C. ciliaris*), Silviculture ($T_5$—*Acacia nilotica*) and Silvi-pasture ($T_6$—*A. nilotica* + *C. Ciliaris*) on soil properties, runoff interception, sediment trapping and soil loss reduction. The lowest average annual soil loss (4.83 ton ha$^{-1}$ year$^{-1}$) and runoff (109.52 mm) were recorded under $T_4$, while the highest sediment loss (8.09 ton ha$^{-1}$ year$^{-1}$) and runoff (136.07 mm), respectively, were under $T_5$. The runoff coefficient of land uses was in the order of $T_3$ (20.30%) < $T_4$ (20.56%) < $T_1$ (21.95%) < $T_2$ (22.26%) < $T_6$ (22.83%) < $T_5$ (25.54%). The *C. ciliaris* improved bulk density and recorded lowest in horti-pasture (1.63 ± 0.04 g cm$^{-3}$) followed by pasture (1.66 ± 0.03 g cm$^{-3}$) land use system. The active SOC content in pasture, horti-pasture and silvi-pasture was 0.95, 0.87 and 0.64 times higher, respectively, than agriculture land use. Under pure *C. ciliaris* cover, resistance to penetration varied from 0.68 to 1.97 MPa, while in silviculture land use, it ranges from 1.19 to 2.90 Mpa. Grass cover had substantial impact on soil loss and runoff reduction, SOC content, soil aggregation and resistance to penetration. In degraded ecosystems, *Cenchrus ciliaris* can be used alone and in combination with plants for protection of natural resources from water-induced soil erosion, runoff conservation, soil quality improvement and maximization of precipitation water use.

**Keywords:** aggregates; *Cenchurus ciliaris*; Chambal river; runoff coefficient; ravine; sediment

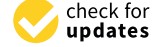



## 1. Introduction

Natural resources are being degraded rapidly worldwide. Land is facing expeditious pressure for various demands along with worldwide land degradation problems [1,2]. Land degradation refers to loss in soil productivity including its present and potential capabilities through deterioration of physical, chemical, biological and socio-economic features. The highest continental percentage of land degradation is in Asia (37%), while the lowest in Central America (3%), the world total being 15% [3]. Oldeman [4] classified that approximately 2 percent of the degraded lands are irreversible degraded, 7 percent are moderately degraded, and 6 percent are lightly degraded. In India, 53% of total geographical area is degraded due to one or more kinds of land degradation. Water-induced soil erosion is the foremost cause of land degradation, affecting about 2 billion ha

(22.5 percent of agricultural land, pasture, forest and woodland) all over the world [3,5–7] comprising 70 percent of all dry lands (about 3.6 billion ha).

The worst forms of land degradation are gullied and ravine land formations. Ravine land is a system of gullies running more or less parallel to each other and then converge into a major river system or its tributary with the formation of riverine terraces (locally known as *Bihad/Kachhars*). Land degradation due to gully erosion is one of the severe threats to the vast tracts of world's agricultural land [8]. The severe form of gully erosion leads to the formation of ravine and, in the most serious cases, the process of ravine formation is almost irreversible. The formation of these severe forms of land degradation starts from river banks, and if not protected, then a network of gullies progressively encroaches the adjoining catchment area [9,10]. The formations of ravine zones of India have no obvious relation to climate. Extreme climatic events, human activities and continued deforestation exposes soil which may exacerbate ravine expansion. Ravine expansion also supported by polycyclic character of river, morphology of the region and neo-tectonics activity in the region [11,12]. The gullied land can be characterized by highly degraded undulating landscapes along with steep multidirectional slops, unscientific land use or land management, scoring the action of rainfall drainage density and activeness [2,13]. The gully formation process and many other factors in ravine as well as in catchment area play a key role in the formation and extension of riparian zones along the major river system all over the world, including India. The National Commission on Agriculture [14] reported 3.67 million ha (1.12% of total geographical area) of ravine land in India (first legitimate estimation). Recently, the Indian Council of Agricultural Research, New Delhi, reported 1.037 million ha ravine land in four major ravine infested states, i.e., Rajasthan, Uttar Pradesh, Madhya Pradesh and Gujarat of India [15].

Many land management technological packages and practices have been developed and recommended by many organizations/institutes/scientists for conservation and utilization of degraded land [16–18]. Appropriate soil and water conservation measures must be adopted to prevent further degradation of marginal land. The mechanical and physical structures, for example, peripheral/or marginal bunds, land leveling and slope smoothing, graded bunding, bench terracing and gully head and gully bed stabilizing by different kind of spill ways (drop spill way, chute and pipe spill way), gully plugs, check dams and bori bunds [19–21] are well-proven means to control soil erosion, but they are cost and energy intensive, and may not be suitable to highly undulating and fragile ravine-like topography. Biological means, particularly grass-based methods, have been reported to be very cost effective and suitable for undulating, unstable and sloppy lands [7]. Land use and land cover are promising factors affecting the runoff and soil loss [22–25]. Perennial grasses such as *Cenchrus ciliaris*, *Pennisetum purpureum*, *Saccharum munja*, *Dichanthium annulatum*, *Thysanolaena maxima*, etc., provide ground cover throughout the year and seem to possess the most desirable attributes as an effective grass barrier for controlling surface runoff and soil loss [26,27]. One of the major components of our study was the inclusion of *C. ciliaris* alone and in combination with selected land use systems for the study.

*C. ciliaris* has high economic and nutritional value and is one among the pastoral species preferred as good quality feed for animals, especially in tropical and subtropical regions [28–31]. Recently, several researchers have reported the effective role of *C. ciliaris* for the rehabilitation of mine spoil areas, ravine areas, degraded land and erosion control [32–34]. *C. ciliaris* was tested for this purpose because it has enough tolerance to moisture deficit conditions, the capacity to withstand heavy grazing, a vast and deep root system and the ability to keep a responsive nature to precipitation [28]. The apomictic [35] nature (capacity to produce clones from seeds) of the *C. ciliaris* grass is the most suitable trait for use in the rehabilitation of degraded lands such as ravine land, mine spoil area, etc. Due to the apomictic, invasive and spreading nature of *C. ciliaris* grass, it can spread in monotype stand, small clumps and clusters throughout the landscape. The strong, wide and deep root system along with clumpy nature of *C. ciliaris* grass intuits an erosion-controlling quality

in it. The root system of a single club can create a hold on a 1.50-to-2.0 m-wide and up to 1.0 m-deep soil system.

Recently, in most of the cases, the studies on the impact of grasses/grass strips on erosion control were carried out in simulated or normal field experiments [36–39]. Very few explained the conservative nature of *C. ciliaris* grass under natural rainfall chronic conditions, especially in ravine lands. To assess the impact of *C. ciliaris* alone and in combination on natural resources conservation, six land use and land covers were tested under semi-arid climatic condition in the fringe zone of the Chambal riparian ecosystem of India. The aim of our experiment was to assess the efficacy of selected land uses to reduce the sediment and runoff losses from selected land uses over the period of nine years (2011–2019) in marginal lands of the Chambal ravines. The experiment was conducted on the piping-prone, swelling and shrinking and erosion-sensitive nature of silty clay loam soils. The long-term period of experiment allowed us to study the temporal variability in rainfall, efficacy of selected land use systems to control erosion and runoff, vegetation development and soil properties state under the natural environment. Initially, we assumed that upon completion of a long-term study, selected land use and land covers will have a definite and strong impact on erosion-controlling soil properties (BD, aggregation, SOC content, resistance to penetration, etc.), sediment and runoff generation. The results of our study are presented and discussed in the forthcoming sections. The results of the study represent the effect of land use and covers on selected soil parameters, runoff and soil loss after completion of a continuous nine year period.

## 2. Material and Methods

### 2.1. Study Area

The study was conducted in semi-arid climatic condition on marginal lands (fringe zone) of Chambal ravine region, located in Kota, Rajasthan state of Western India, having latitude 25°13′51.30″ N, longitude 75°52′22.51″ E, and about 254 m a.s.l.) (Figure 1). Mean annual rainfall and temperature were 741 mm and 25.8 °C, respectively. The mean winter season temperature (MWST) and mean summer season temperature (MSST) are greater than 5 °C. The area qualifies for *ustic* soil moisture regime and hyperthermic temperature regime. More than 90% rainfall is received during July to September months from south-western monsoon. Remaining part of the year remains in water deficit with mean annual potential evapotranspiration demand of 1036.60 mm/year. Soil texture varied from sandy clay loam to clay through sandy loam and sandy clay with slow-to-moderate drainage capability. Geologically, these soils originated from sandstone, shale and limestone [40]. Presence of calcium carbonate concretions throughout profile is a common phenomenon in the region. The soils have a swelling and shrinking nature which encourages a high amount of surface flow. The organic carbon content of soil varies between low and medium class, i.e., 0.20 to 0.45% [41,42].

### 2.2. Experimental Setup

To assess the impact and efficacy of predominant land use systems of semi-arid climatic conditions of south-eastern Rajasthan on sediment loss and runoff under natural environmental conditions, a long-term study (2011–2019) was conducted at ICAR-Indian Institute of Soil and Water Conservation, Research Center, Kota, Rajasthan, Western India. The study was conducted on Typic Haplusterts (medium and deep black) soils [43] highly susceptible to swelling, shrinking, cracking and hardening. A total of six land use systems were tested for their suitability in terms of permissible runoff and sediment yields in Chambal ravine region soils of south-eastern Rajasthan. To overcome the impact of aspect and elevation gradient on precipitation regime, vegetation cover and spatially distributed radiation [44], experimental site was selected with similar aspect (north-east facing) and straight slope to implement selected land use system. The selected land use systems (Figure 2) were: Agriculture (Rainfed Soybean)—$T_1$; Agri-horticulture (Soybean+ *Manilkara achras*)—$T_2$; Horti-pastoral (*Emblica officinalis + Cenchrus ciliaris*)—$T_3$; Pasture (*Cenchrus ciliaris*)—$T_4$;

Silviculture (*Acacia nilotica*)—$T_5$; and Silvi-pasture (*Acacia nilotica + Cenchrus ciliaris*)—$T_6$. At the outset (in 2010), experimental site was prepared by clearing and removing the sparse vegetation dominated by *Prosopis juliflora* and *Leucenia leucocephala.* Plot homogeneity and design of experiments was ensured before implementation of selected LUSs. Six plots of 30 m length × 45 m width (each plot is subdivided in three part of 30 m length × 15 m width for sediment loss and runoff estimation) were established at experiment site. A drainage channel of 3.0 m top width, 1.0 m bottom width and 1.0 m depths had been excavated for removing excess water from the experimental area and safe disposal of collected water from runoff collection tank (1.4 × 0.70 × 0.70 m dimension).

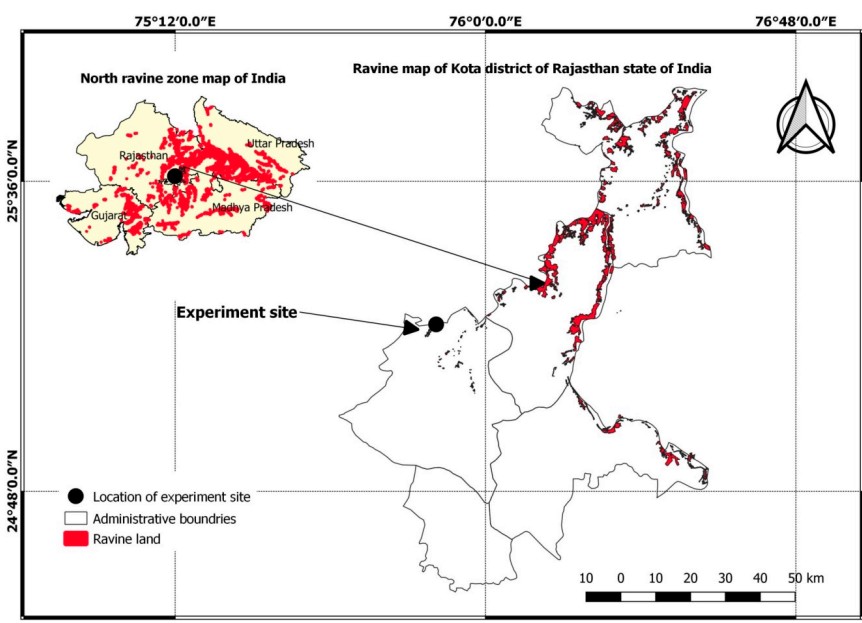

**Figure 1.** The geographical location of the experimental site in Kota district, Rajasthan, India.

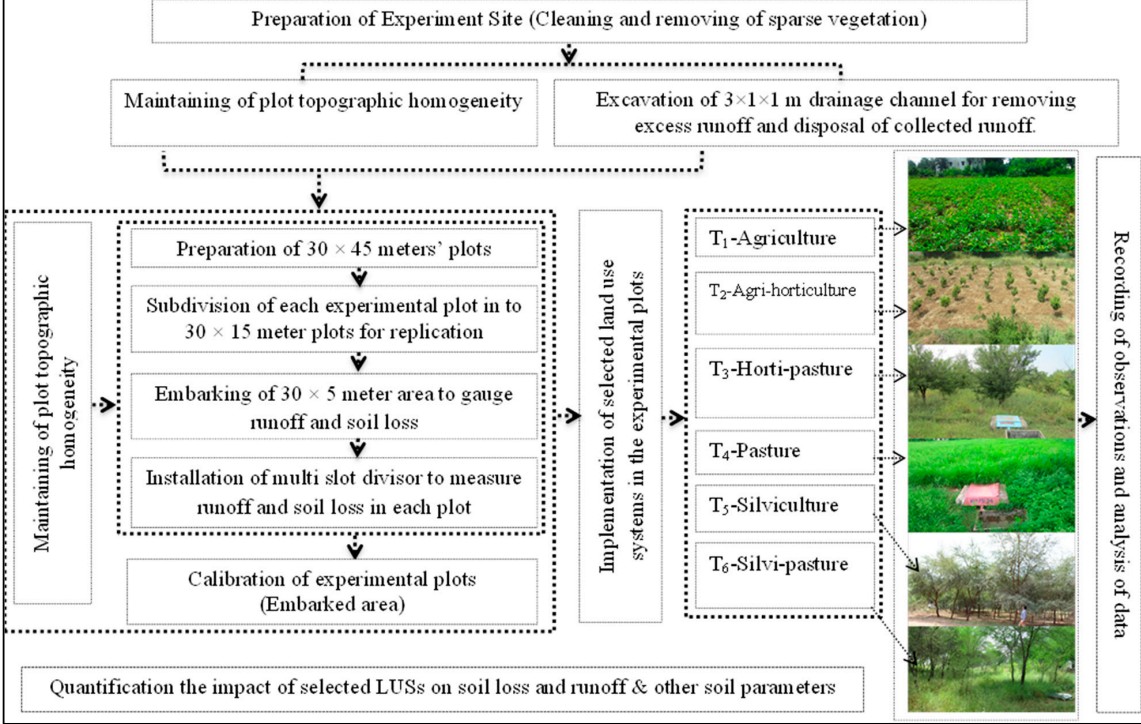

**Figure 2.** Flow chart of methodology and layout details with field view of one replication under different land use systems.

### 2.3. Runoff and Soil Loss Estimation

Eighteen multi-slot divisor (having nine slots) were fixed at lower sides of experimental plots and used for monitoring runoff and soil loss. Out of the total experimental area of $15 \times 30$ m in each plot, $5 \times 30$ m plot was gauged by multi-slot divisor. Recording-type rain gauge was used to gauge precipitation in the experimental area. The year 2010–2011 was kept for calibration and runoff and sediment loss from all experimental plots. Calibration of the instrument (multi slot divisor) was performed by adopting standard process [45,46]. The experimental plots were also calibrated to ensure consistent quantitative estimate of the variations in runoff behavior of the plots. The runoff was measured after each rainfall event by measuring the depth of collected runoff in runoff collection tank. After runoff estimation, the collected water in tank was well harmonized to thorough mix the lost sediment and then 0.50 L water sample was collected to measure soil loss in each event. Collected runoff samples were later transferred into beaker and evaporated at 105 °C to assess the event wise soil loss. The runoff coefficient (%) was determined as the ratio of runoff (mm) to the seasonal precipitation (mm).

### 2.4. Plantation, Sodding of Grass and Crop Cultivation

All the land use treatments were imposed after calibration of multi-slot divisors and runoff plots. Pits of size $1 \times 1 \times 1$ cm were dug out at desired spacing during the month of April to May. The dugout soil was left for desiccation during May–June. Seedlings (at least one year old) of the trees were planted after onset of first monsoon rains. Plant-to-plant distance was kept at $5 \times 5$ m for plantation of horticulture plants (*Manilkara achras* and *Emblica officinalis*) and silvicultural plant (*Acacia nilotica*). The interspaces were planted with *C. ciliaris* grass at $30 \times 30$ cm spacing by making a furrow or pit of 15 cm depth across the straight land slope (with nearly level gradient of 0.5–1.0%). For a single pit, four to five tillers are sufficient enough to be sodded to reduce soil erosion hazards. In $T_2$ treatment, soybean (*Glycin max*) was taken along with *Manilkara achras* to observe the effect of crop cultivation and tillage on soil loss and runoff. Sowing was performed across the slope of the field to promote in situ moisture conservation and reduce soil erosion. In treatments $T_1$ and $T_2$, soybean crop was grown annually, and standard package of practices were followed. The grass production increases from the 2nd season to 5th season of harvest and declined gradually. Hence, about 1/3rd area was re-sodded at the end of every fifth year for sustained grass production from the area.

### 2.5. Physicochemical Properties of Soil

The important soil properties such as texture, bulk density, resistance to penetration, organic matter, soil aggregates, etc., and land use/cover are considered important in erosional and hydrological processes. To assess the influence and role of these parameters, soil sampling was performed from four depths (0–15, 15–30, 30–45 and 45–60 cm) in each land use. Three soil samples for each soil depth from three different locations for every land use system were collected. Thus, 12 soil samples for every land use system and 72 soil samples from whole experimental site were collected and moved to the laboratory. The samples were air dried and gently crushed with a wooden mortar pastel and passed through 2 mm sieve. A part of soil samples (sub-sample) was grounded to pass through 0.5 mm sieve. Samples were treated with 35% $H_2O_2$ to destroy organic matter and with calgon (sodium hexametaphosphate) to disperse them for particle size analysis by the international pipette method [47]. The pH and EC of soil were determined by using 1:2 soil and water suspension and 1:2 soil and water extract using pH and EC meter (by Systronics), respectively [48]. Organic carbon (OC) was determined by titration method [49]. Bulk density was measured thorough core sampler of known volume. Soil resistance was assessed using penetrologger of Eijlkelkamp. The available micronutrients N, P, and K were estimated by alkaline permanganate [50], $SnCl_2$ reduced phosphomolybdate [51], flame photometer, respectively. The cationic micronutrients $Fe^{2+}$, $Mn^{2+}$, $Zn^{2+}$ and $Cu^{2+}$ were determined through DTPA extraction [52] methods.

### 2.6. Estimation of Active Soil Organic Carbon (Active SOC)

The labile fraction of soil C pool, often termed as the active SOC pool, serves as sensitive indicator of changes in management-induced soil quality [53]. Land degradation or improvement in response to land uses and/or management practices can easily be presumed by slight variation in active SOC. The active SOC was determined by oxidation of labile SOC by $KmnO_4$ (0.01 M strength). The pH of $KmnO_4$ solution was adjusted to 7.2 with 0.1 M NaOH. To estimate the amount of C oxidized, it is reported that 1 mol $MnO_4^-$ is consumed (reduced from $Mn^{7+}$ to $Mn^{4+}$) in the oxidation of 0.75 mol (9000 mg) of C [54]. The bleaching of the purple $KmnO_4$ color is proportional to the amount of oxidizable carbon in soil.

Five grams of air-dried soil was taken in a 150 mL conical flask. An amount of 20 mL 0.01 M $KmnO_4$ solution was added to it, followed by 0.3 g calcium chloride addition to enhance settling of soil. The suspension was shaken at 200 rpm for 5 min. After shaking, the suspension was centrifuged at 3000 rpm for 5 min and filtered through glass fiber filters. The bleaching of color of $KmnO_4$ was measured by spectrophotometer at 550 nm light setting. The chemical reaction during the oxidation of carbon is:

$$3C + 4KmnO_4 + 2H_2O = 4KOH + 4MnO_2 + 3CO_2 \tag{1}$$

Active soil organic carbon was calculated as:

$$\text{Active C (mg/kg)} = [0.01 \text{ mol/L} - (a + b \times \text{absorbance})] \times (9000 \text{ mg C/mol}) \times (0.02 \text{ L solution}/0.005 \text{ kg soil}) \tag{2}$$

where 0.01 mol/L is the initial concentration of $KmnO_4$, 'a' is the intercept and 'b' is the slope of the standard curve. The 0.005 is the amount of soil in kg on oven dry basis.

### 2.7. Mean Weight Diameter (MWD)

For expressing the distribution of aggregate sizes, mean weight diameter (MWD) was used to integrate aggregate size distributions obtained by mechanical sieving. Mean weight diameter (MWD) was determined by Yoder's apparatus wet sieving method [55]. For this, 100 g air-dried, 4–8 mm size range aggregate samples were used. The wet sieving of the air-dried soil sample was carried out using a nest of sieves with mesh openings of 4.75, 2.00, 1.00, 0.50, 0.25, 0.10, 0.05 and <0.005 mm, respectively. The distributed soil aggregates were collected separately in each sieve and weighed to compute different soil aggregate classes with respect to the total soil sample weight. The size distribution of aggregates was characterized by mean weight diameter [56] by using the following formula;

$$\text{MWD} = \sum_{i=1}^{n} \overline{Xi} \, Wi \tag{3}$$

where MWD represents mean weight diameter. *Xi* is the mean diameter of aggregates separated by sieving and *Wi* is the proportion of the total dry sample mass, and the summation was carried out to overall n size fractions.

### 2.8. Statistical Analysis

In the present study, the variation in soil properties, means and standard deviations were computed in MS Excel (Microsoft Office Professional Plus 2010 version). The impact of different land use systems was assessed through impact on soil loss, runoff, MWD, mean values and standard errors, analysis of variance (one-way ANOVA), and significant differences among data were determined at $p = 0.05$ significance level. The fine particles (silt + clay) content was selected as the index property and regressed against pH in order to gauge the impact of fine particles on soil pH.

## 3. Results and Discussion

### 3.1. Physico-Chemical Properties of Soil

The soil texture of the study area varied from sandy clay loam to clay through sandy loam and sandy clay (Table 1). The finer fraction (silt plus clay) constitutes 50–70% of whole textural composition of soil. A specific trend of soil textural composition variation from upper to lower layers was not observed under all land use systems. The bulk density varied from $1.63 \pm 0.04$ to $1.78 \pm 0.04$ g cm$^{-3}$. The lower bulk density was recorded in horti-pasture ($1.63 \pm 0.04$ g cm$^{-3}$) and *Pasture* ($1.66 \pm 0.03$ g cm$^{-3}$) land uses, while higher ($1.72 \pm 0.02$ g cm$^{-3}$) was in silviculture and silvi-pasture ($1.78 \pm 0.04$ g cm$^{-3}$) land uses. Both land use and soil depth had a significant effect on bulk density at $p \leq 0.05$. Overall, as depth proceeds, the bulk density increases from the top to the lower layer under all selected land uses. The results reveal that under treatment $T_3$ and $T_4$, the soil bulk density improves more than other land uses over the period. Hence, the notion that grasses improve the soil properties [57,58] proved true in our study after completion of nine years of experiment. Before plantation, the study site was occupied by scattered and busy vegetation. It was observed that under treatment $T_5$ and $T_6$, the values of bulk density and resistance to penetration were higher than other land uses along with generation of a higher amount of mean annual runoff and soil loss. Thus, our results coincide with the findings of Dominy et al. [59], Rasiah et al. [60], Sharrow [61] and Geissen et al. [62], who reported that lands' conversion to introduce any new and permanent perennial nature land use, especially plantation, during initial years leads to severe compaction and negative impact on soil structure. In our study, the higher amount of runoff from tree land uses also supports land conversion to plantation, with the converse impact on soil bulk density and soil resistance to penetration.

**Table 1.** Physico-chemical soil properties under different land use systems.

| Land Use | Soil Depth (cm) | Soil Texture | | | Bulk Density (g·cm$^{-3}$) | Organic Carbon (%) | Texture Class |
|---|---|---|---|---|---|---|---|
| | | Sand (0.05–2.0 mm) | Silt (0.05–0.002 mm) | Clay (<0.002 mm) | | | |
| Soybean Crop | 0–15 | 56.0 | 12.0 | 32.0 | 1.696 | 0.47 | scl * |
| | 15–30 | 58.0 | 12.0 | 30.0 | 1.712 | 0.40 | scl |
| | 30–45 | 56.0 | 14.0 | 30.0 | 1.699 | 0.30 | scl |
| | 45–60 | 46.0 | 16.0 | 38.0 | 1.747 | 0.27 | sc ** |
| | Mean ± SE | 54.00 ± 2.71 | 13.50 ± 0.96 | 32.50 ± 1.89 | 1.71 ± 0.01 | 0.36 ± 0.05 | |
| | SD | 5.42 | 1.91 | 3.79 | 0.02 | 0.09 | |
| Agri-Horticulture | 0–15 | 56.0 | 16.0 | 28.0 | 1.610 | 0.43 | scl |
| | 15–30 | 42.0 | 20.0 | 38.0 | 1.612 | 0.27 | CL *** |
| | 30–45 | 64.0 | 20.0 | 16.0 | 1.737 | 0.27 | SL **** |
| | 45–60 | 44.0 | 16.0 | 40.0 | 1.725 | 0.24 | c ***** |
| | Mean ± SE | 51.50 ± 5.19 | 18.00 ± 1.15 | 30.50 ± 5.50 | 1.67 ± 0.03 | 0.30 ± 0.04 | |
| | SD | 10.38 | 2.31 | 11.00 | 0.07 | 0.28 | |
| Horti-Pasture | 0–15 | 38.0 | 18.0 | 44.0 | 1.573 | 0.45 | c |
| | 15–30 | 40.0 | 20.0 | 40.0 | 1.543 | 0.38 | c |
| | 30–45 | 36.0 | 20.0 | 44.0 | 1.659 | 0.27 | c |
| | 45–60 | 34.0 | 18.0 | 48.0 | 1.737 | 0.21 | c |
| | Mean ± SE | 37.00 ± 1.29 | 19.00 ± 0.58 | 44.00 ± 1.63 | 1.63 ± 0.04 | 0.33 ± 0.05 | |
| | SD | 2.58 | 1.15 | 3.27 | 0.09 | 0.11 | |
| Pasture | 0–15 | 48.0 | 18.0 | 34.0 | 1.619 | 0.56 | scl |
| | 15–30 | 32.0 | 22.0 | 46.0 | 1.636 | 0.32 | c |
| | 30–45 | 42.0 | 18.0 | 40.0 | 1.756 | 0.23 | c |
| | 45–60 | 34.0 | 20.0 | 46.0 | 1.612 | 0.17 | c |
| | Mean ± SE | 39.00 ± 3.70 | 19.50 ± 0.96 | 41.50 ± 2.87 | 1.66 ± 0.03 | 0.32 ± 0.09 | |
| | SD | 7.39 | 1.91 | 5.74 | 0.07 | 0.17 | |
| Silviculture | 0–15 | 48.0 | 18.0 | 34.0 | 1.759 | 0.53 | scl |
| | 15–30 | 46.0 | 18.0 | 36.0 | 1.720 | 0.44 | sc |
| | 30–45 | 52.0 | 18.0 | 30.0 | 1.773 | 0.60 | scl |
| | 45–60 | 54.0 | 16.0 | 30.0 | 1.695 | 0.53 | scl |
| | Mean ± SE | 50.00 ± 1.83 | 17.50 ± 0.50 | 32.50 ± 1.50 | 1.74 ± 0.02 | 0.52 ± 0.03 | |
| | SD | 3.65 | 1.00 | 3.00 | 0.04 | 0.07 | |

**Table 1.** *Cont.*

| Land Use | Soil Depth (cm) | Soil Texture | | | Bulk Density (g·cm⁻³) | Organic Carbon (%) | Texture Class |
| --- | --- | --- | --- | --- | --- | --- | --- |
| | | Sand (0.05–2.0 mm) | Silt (0.05–0.002 mm) | Clay (<0.002 mm) | | | |
| Silvi-pasture | 0–15 | 58.0 | 12.0 | 30.0 | 1.801 | 0.47 | scl |
| | 15–30 | 52.0 | 16.0 | 32.0 | 1.854 | 0.44 | scl |
| | 30–45 | 52.0 | 12.0 | 36.0 | 1.658 | 0.41 | sc |
| | 45–60 | 50.0 | 14.0 | 36.0 | 1.789 | 0.30 | sc |
| | Mean ± SE | 53.00 ± 1.73 | 13.50 ± 0.96 | 33.50 ± 1.50 | 1.78 ± 0.04 | 0.40 ± 0.04 | |
| | SD | 3.46 | 1.91 | 3.00 | 0.08 | 0.20 | |

* Silty clay loam, ** Sandy clay, *** Clay loam, **** Sandy loam, ***** Clay.

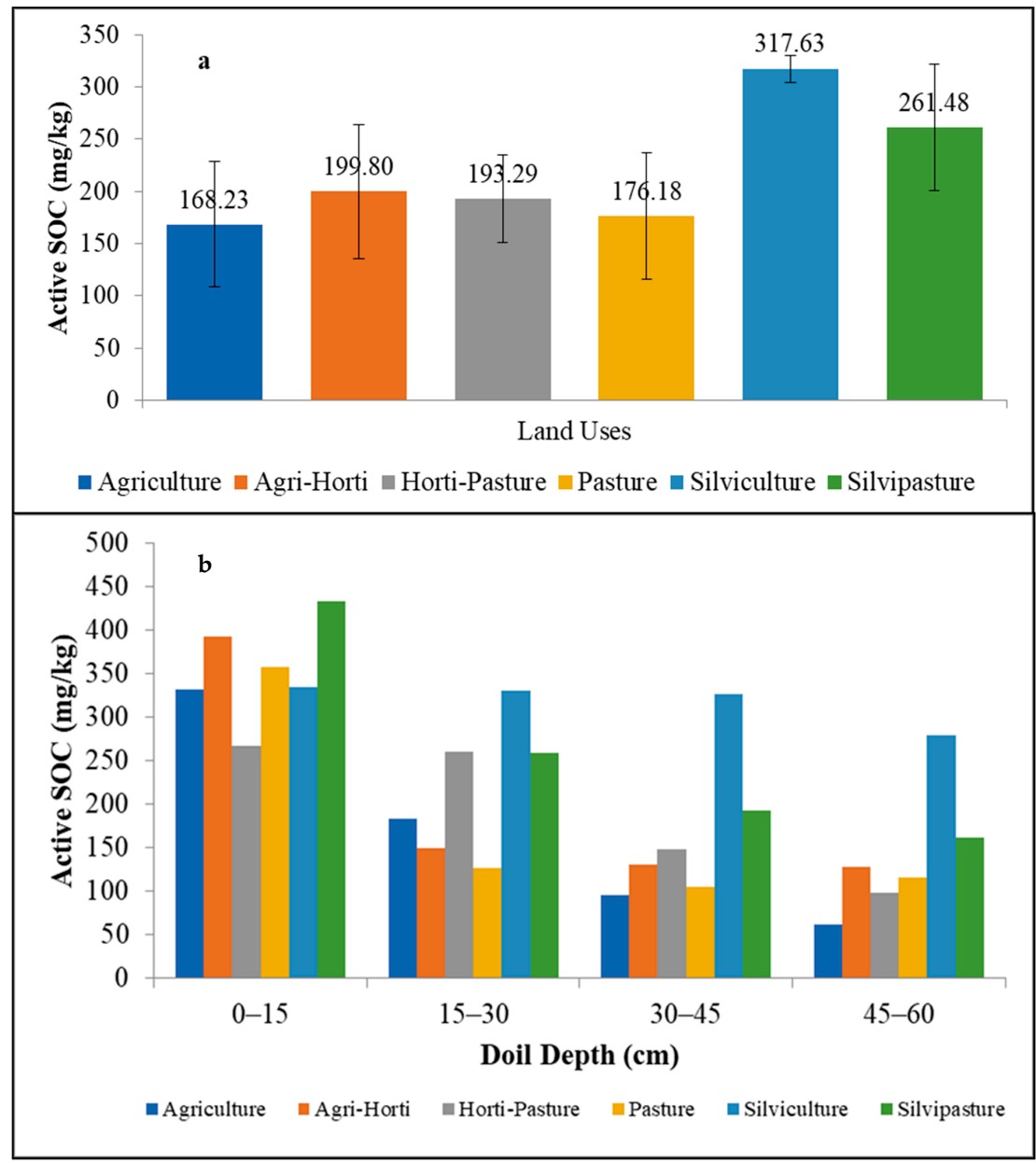

**Figure 3.** Variation in active soil organic carbon (mg kg⁻¹) (**a**) in whole soil profile (0–60 cm soil depth) and (**b**) in different soil layers under semi-arid climatic condition. Error bar (**a**) indicates the standard error values of active SOC under different land uses. $T_1$—Agriculture, *Glycin max.*; $T_2$—Agri-Horti, *Manilkara achras* and *Glycin max*; $T_3$—Horti-Pasture, *Emblica officinalis* and *Cenchrus ciliaris*, $T_4$—Pasture, *Cenchrus ciliaris*; $T_5$—Silviculture, *Accacia nilotica*; $T_6$—Silvipasture, *Accacia nilotica* and *Cenchrus ciliaris*.

Although the soil organic carbon (Table 1) and oxidizable/active SOC content (Figure 3) in sub-tropical soil does not significantly vary, it was observed that plantation land use and its combination with grass had a distinct advantage in maintaining the higher amount of both fractions of SOC compared to cultivated soil. In the present study, it was found that the plough layer under all land uses had a higher amount of SOC than the lower layer. The active SOC content in pasture, horti-pasture and silvi-pasture was 0.95, 0.87 and 0.64 times higher, respectively, than that of the cultivated field. The silviculture land uses contributed SOC throughout the whole soil profile. The higher amount of SOC fractions under grasses and silviculture land uses could be attributed to the continuous addition of leaf litter and root decay over the period of time. The addition and comparatively high amount of SOC in the top 15 cm soil layer under all land uses leads to a positive impact on bulk density and increased porosity [63]. The higher labile SOC in plantation land uses could be attributed to the fact that when soil compaction and bulk density increases, they ultimately lead to more root biomass (almost twofold higher) in per unit area as compared un-compacted sites [64].

*3.2. Resistance to Penetration*

Resistance against penetration, a characteristic of soil compaction, did not cross the optimal soil resistance limit for healthy and undisturbed root growth, but it showed a non-significant difference among the selected land use systems up to 0–15 cm soil depth (Figure 4). The optimal resistance to penetration in tropical soils varies from 1.0 Mpa to 3.0 Mpa [65]. In the present study, minimum resistance to penetration (0.68 to 1.97 Mpa) was observed under *C. ciliaris* land use, while under silviculture land uses, it varies from 1.19 to 2.90 Mpa (Figure 3). As depth proceeds, the resistance to penetration also increased from the upper to lower soil layers. In treatment $T_6$ (silvi-pasture), the growth of *C. ciliaris* grass was gradually hampered by the shadow along with the phytogeneric impact of *Acacia nilotica* trees, and upon 5–6 years of experiment, *C. ciliaris* grass went extinct from the experimental field. As a result, both $T_5$ and $T_6$ treatments behave in a similar way from the 6th year of experiment onward. Both plantation land uses lead to more compaction compared to pasture/horti-pasture land uses. The results were also supported by the increased bulk density under these land uses. The results of the study were in line with findings by Rasiah et al. [42] and Geissen et al. [44], who reported that permanent land uses leads to severe soil compaction and increased bulk density. Comparatively increased values of resistance against penetration in tree sites were observed to be more than three times higher. This leads to the conclusion that land conversion to introduce permanent land uses, especially the plantation of trees, initially leads to decreasing soil quality due to soil compaction. Recently, it has been reported by many researchers [66–68] that soil properties vary significantly between plantations and natural forests. Some soil properties such as soil pH, SOC content and plant available nutrients reduce upon land conversion to plantation, but at the same time, some important soil physical properties such as bulk density, soil compaction, etc., are higher in plantations than in natural forests [69,70].

The soils under selected land uses did not significantly vary with respect to pH and soluble salt concentration in soil solution (Table 2). The lowest pH of soils in treatment $T_1$ was $7.83 \pm 0.05$, while in treatment $T_4$, the highest pH was $8.55 \pm 0.16$. The soil pH in most cases increased with depth in all profiles. The deposition of $CaCO_3$ concretion was noticed in lower horizons in the experimental area. The downward movement of soluble salts, deposition of illuviated bases [71–73] increased $CaCO_3$ content in subsurface horizons and the semi-arid climatic condition of the region could be attributed to the higher range of soil reaction. The proportion of fine particles (clay and silt) also showed a significant influence on soil reaction, conjointly governing 49.7% variations in soil reaction (Figure 5). The average proportion of the fine particles in soil profile (0–60 cm) was used to correlate it with the soil reaction. The results showed that the neither the slope nor the intercept differed significantly, respectively, from the unity and zero at probability level, $p = 0.0001$, for the pH and fine particles in soil. This validated that derived calibration prediction equations were useful for analyzing the effects of the fine particles on soil reaction. The electrical

conductivity (EC) of the soil under all land uses was low and varied from $0.20 \pm 0.01$ ($T_2$—Agri-horticulture) to $0.33 \pm 0.11$ ($T_4$—Pasture) dSm$^{-1}$ indicating no salinity problem in these soils. Even though higher soluble salt content was observed in the upper layer than deeper soil horizons under all land uses, which could be attributed to the semi-arid climatic conditions, high potential evepo-transpiration demand of the region and the movement of soluble salts to upper layer, which had a bearing of localized conditions on their movement in the profiles [74,75].

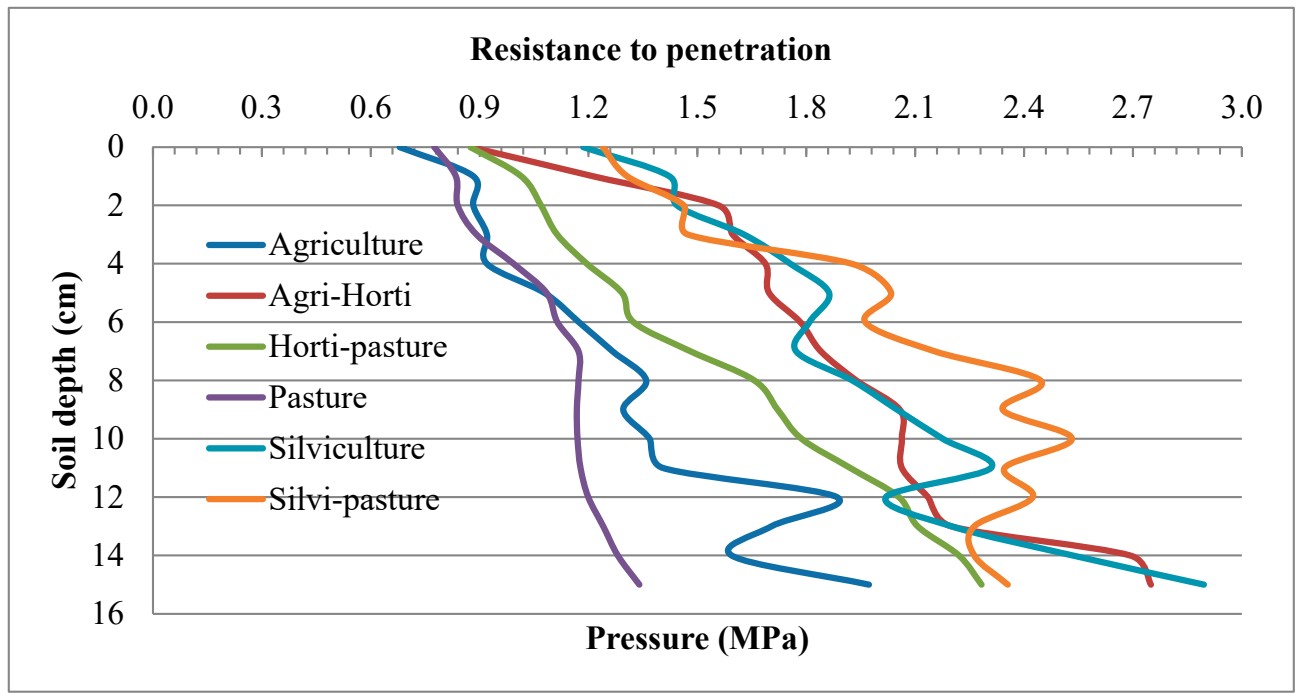

**Figure 4.** Resistance to penetration under different land use systems in semiarid arid climatic condition. $T_1$—Agriculture, *Glycin max.*; $T_2$—Agri-Horti, *Manilkara achras* and *Glycin max*; $T_3$—Horti-Pasture, *Emblica officinalis* and *Cenchrus ciliaris*; $T_4$—Pasture, *Cenchrus ciliaris*; $T_5$—Silviculture, *Accacia nilotica*; $T_6$—Silvipasture, *Accacia nilotica* and *Cenchrus ciliaris*.

**Table 2.** Annual, seasonal and runoff generating rainfall (mm) and runoff producing events during the experimental period (2011–2019).

| Years | Annual Rainfall (mm) | Seasonal Rainfall (mm) | Runoff Generating Rainfall (mm) | Monsoon Season Rainfall Events (Nos.) | Runoff Generating Events (Nos.) | Deviation from Normal Monsoon Rainfall, i.e., 675.10 mm (%) |
|---|---|---|---|---|---|---|
| 2011 | 1066.50 | 1029.40 | 885.10 | 36 | 17 | 31.11 |
| 2012 | 782.30 | 731.70 | 462.00 | 35 | 12 | −31.57 |
| 2013 | 1038.60 | 987.80 | 674.40 | 57 | 22 | −0.10 |
| 2014 | 703.70 | 636.20 | 478.32 | 32 | 10 | −29.15 |
| 2015 | 803.70 | 644.20 | 378.40 | 30 | 8 | −43.95 |
| 2016 | 1021.00 | 988.00 | 846.20 | 37 | 24 | 25.34 |
| 2017 | 495.40 | 463.00 | 291.00 | 27 | 6 | −56.90 |
| 2018 | 798.00 | 798.00 | 340.90 | 42 | 15 | −49.50 |
| 2019 | 1415.20 | 1368.60 | 719.60 | 53 | 15 | 6.59 |
| Mean ± SD | 902.71 ± 265.21 | 849.66 ± 272.42 | 563.99 ± 222.38 | 38.78 ± 10.20 | 14.33 ± 6.06 | 16.46 ± 32.94 |
| Std. Error | 88.40 | 90.81 | 74.13 | 3.40 | 2.02 | 10.98 |

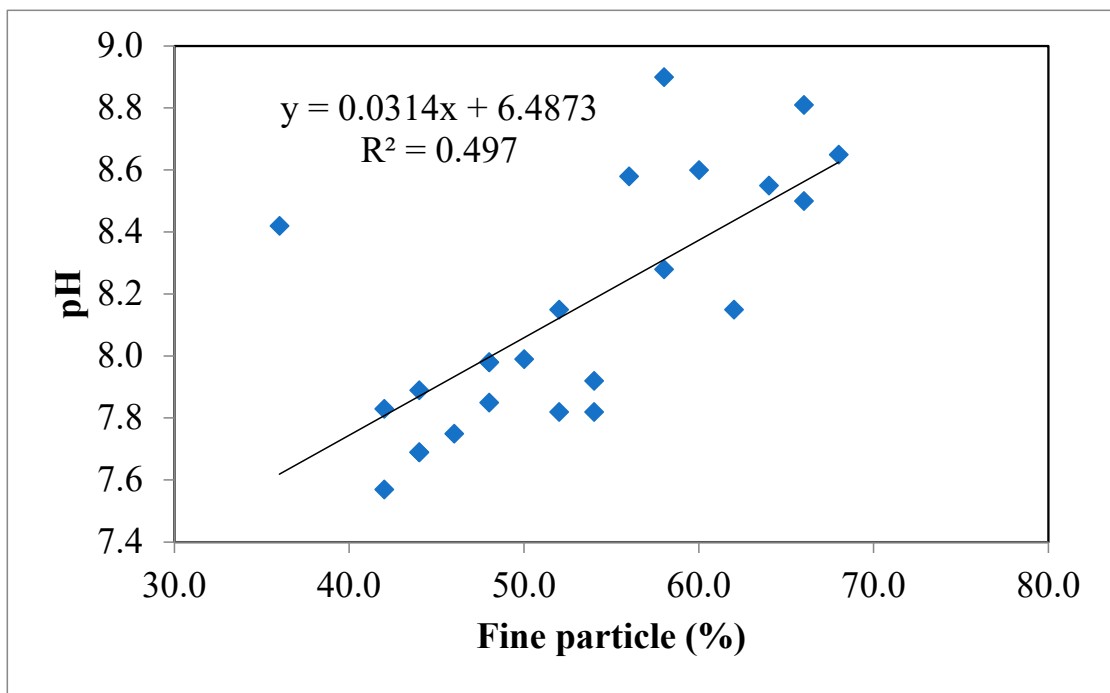

**Figure 5.** Relationship of fine particle (clay plus silt) with soil pH under different land use systems in semi-arid climatic conditions.

*3.3. Mean Weight Diameter*

Soil aggregate stability is a key factor for controlling soil losses and runoff. It improves soil quality, particularly in a degraded ecosystem such as the riparian zone of the Chambal ravine. Soil aggregation is considered as a reflection of soil structure and texture, because the combined group of soil particles is always stronger rather than a single grain [76]. Aggregate stability has a prominently multi-parameter effect on the soil properties. The results of mean weight diameter (MWD) are presented in Figure 6. The significant difference between the different land use systems was observed. The maximum MWD was observed under horti-pasture land use ($T_3$) followed by pasture ($T_4$) land use and plantation land uses, i.e., $T_5$ and $T_6$. The cultivated land use systems ($T_1$ and $T_2$) registered the lowest MWD among all land use systems. The distribution pattern of MWD in soil under selected land use systems supports the notion that vegetation cover has a positive and constructive impact on soil aggregation and soil structure. The lower MWD under crop land uses in our study coincide with the findings of Six et al. [77], Pinheiro et al. [78], Abid and Lal [79] and Mohanty et al. [80], which described that disruption of soil aggregates could be affected by human activity, animals and heavy farm machineries. In the present study, two contrasting facts were observed. One is that the permanent land use systems, especially *Accacia nilotica* plantation ($T_5$ and $T_6$), lead to the compaction and increased bulk density. On the other hand, grass permanent land uses ($T_3$ and $T_4$) enhances soil aggregates stability as indicated by MWD (Figure 6). So, as already stated, conclusively, it can be said that land conversion for plantation may lead to an increase in the stability of aggregates compared to tilled fields, but it also augments soil compaction. In natural forest, this phenomenon may differ from our findings. The inclusion of *C. ciliaris* in land management enhances overall soil quality and fertility. We postulate that increased soil aggregate stability along with increased compactness and bulk density under plantation might be due to the formation of a massive or stratified soil matrix in place of an angular and blocky soil structure. However, this fact is not covered under our study, and it needs further research to confirm and justify the impact on soil structure patterns. The high percentage of silt and clay along with the presence of $CaCO_3$ concretions in the soil matrix may also aggravate the soil resistance.

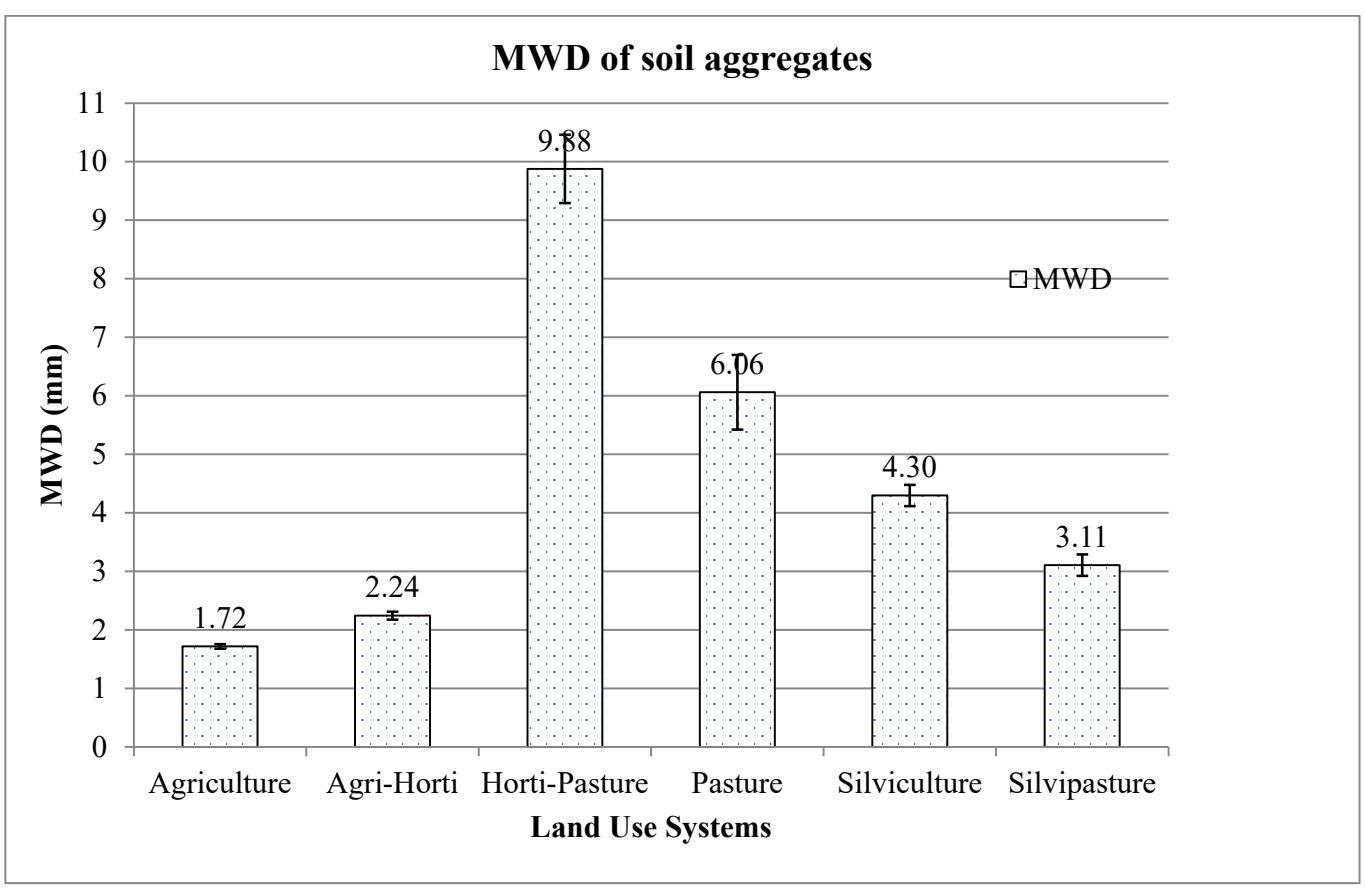

**Figure 6.** Mean weight diameter (MWD) of soil aggregates under different land use systems, which shows significant impact on soil aggregation at $p < 0.05$. $T_1$—Agriculture, *Glycin max.*; $T_2$—Agri-Horti, *Manilkara achras* and *Glycin max*; $T_3$—Horti-Pasture, *Emblica officinalis* and *Cenchrus ciliaris*; $T_4$—Pasture, *Cenchrus ciliaris*; $T_5$—Silviculture, *Accacia nilotica*; $T_6$—Silvipasture, *Accacia nilotica* and *Cenchrus ciliaris*.

*3.4. Runoff, Soil Loss and Available Plant Nutrients*

In the present study, the highest average runoff (136.07 mm) was recorded under a single *Accacia nilotica* plantation, which consequently leads to comparatively more soil loss (8.09 ton ha$^{-1}$ year$^{-1}$) than under other selected land use systems (Figure 7). On an average basis, the cultivated land use systems ($T_1$ and $T_2$) and pasture land use systems ($T_3$ and $T_4$) behave in almost the similar manner to each other throughout the experimental period (2011–2019). The lowest average sediment loss as well as runoff over the nine years of experiment was recorded under treatments $T_4$—pasture land use (4.83 ton ha$^{-1}$ year$^{-1}$ and 109.52 mm)—and $T_3$—horti-pasture land use (5.07 ton ha$^{-1}$ year$^{-1}$ and 108.14 mm)—respectively. The results of our study during the course of the experiment are supported by the findings of García-Ruiz et al. [81], Collins et al. [82], Evans [83], Wang et al. [84] and Nunes et al. [25], who described that the erosion was more serious in arable land and afforested land than in pasture land and other land uses. As mentioned earlier, the area receives most of its rainfall from south-western monsoon during June to September, and the pattern of rainfall distribution over the monsoon period (near about four months) has a very erratic nature. The data in Figure 8 and Table 2 clearly revealed that, on an average basis, the annual rainfall in the season can vary from 495.40 mm to 1415.20 mm with ±265.10 mm variation, while the seasonal rainfall can vary from 463.0 mm to 1368.60 mm with ±272.42 mm variation. The runoff generating rainfall and runoff generating events also showed a variation of ±222.38 mm and ±6.06 with average value of 563.99 mm and 14.33 numbers, respectively (Table 2). Moreover, sometimes the seasonal rainfall of the region completed within two months, and sometimes it was well distributed over the whole four-month monsoon period. This erratic nature of regional rainfall had a direct influence

on runoff and soil loss under selected land use systems (Figures 7 and 8). The highest runoff (mm) was recorded in the year 2019 under all land use systems, while the lowest was recorded in the year 2017 (Figure 8). During 2011–2014, the runoff generated from all selected land uses showed a smooth pattern. On the other hand, data in Figure 9 showed that during 2014, the highest sediment was generated from all land uses. Additionally, in 2018 and 2019, it was very low from all treatments. The erratic and unpredicted nature of the regional rainfall governs the year-to-year variation in runoff and soil loss. The findings are consistent with the findings of Langbein and Schumm, 1985 [85], who reported that vegetation and precipitation exert a competing effect on sediment yield. However, a nonlinear relationship [86] of precipitation–vegetation–erosion could not exactly be stated from the present study. However, on the basis of our findings, we concluded that perennial vegetation, especially coupled with grass cover, had a competing impact with precipitation on erosion and runoff (Figure 7). In our study, to tackle the aspect and elevation gradient impact on the precipitation regime, vegetation cover and spatially distributed radiation [44,87] we carefully implemented LUSs on a north-east facing aspect with a nearly level gradient site. On an average basis, it was found that *Cenchrus ciliaris* had a potential impact on soil loss and runoff reduction. It significantly reduces sediment loss and controls soil erosion. It also reduces runoff generation, which results in more rain water entering into in the soil profile, as evidenced by the reduced runoff and runoff coefficient (Figures 7 and 10). Thus, it can be introduced for soil and water conservation, especially in set-aside ravine-like ecosystems in semi-arid climatic conditions. These findings underline the significance of scrutinizing the coevolution of landforms and vegetation to expand our understanding of the land use system impact on soil and water conservation in degraded ecosystems in natural conditions.

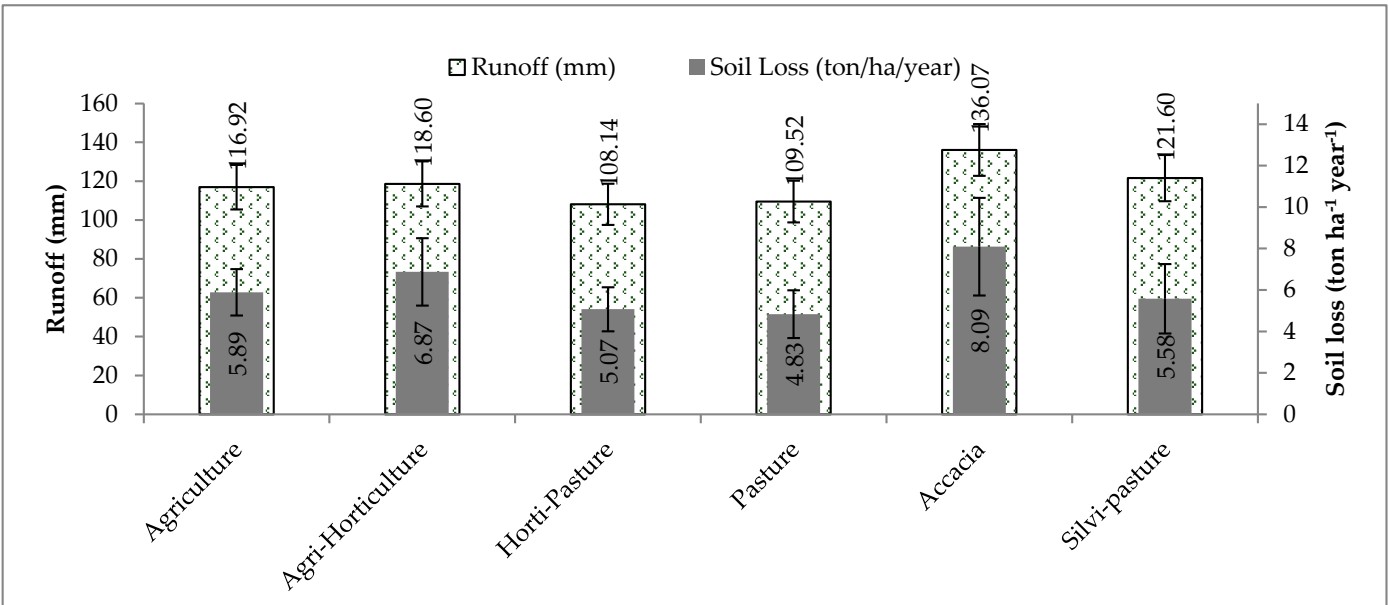

**Figure 7.** Average runoff (mm) and soil loss (ton ha$^{-1}$ year$^{-1}$) under selected land use systems over the period of 9 years of experiment (2011–2019). The error bar denotes ±1 SE. $T_1$—Agriculture, *Glycin max*.; $T_2$—Agri-Horti, *Manilkara achras* and *Glycin max*; $T_3$—Horti-Pasture, *Emblica officinalis* and *Cenchrus ciliaris*; $T_4$—Pasture, *Cenchrus ciliaris*; $T_5$—Silviculture, *Accacia nilotica*; $T_6$—Silvipasture, *Accacia nilotica* and *Cenchrus ciliaris*.

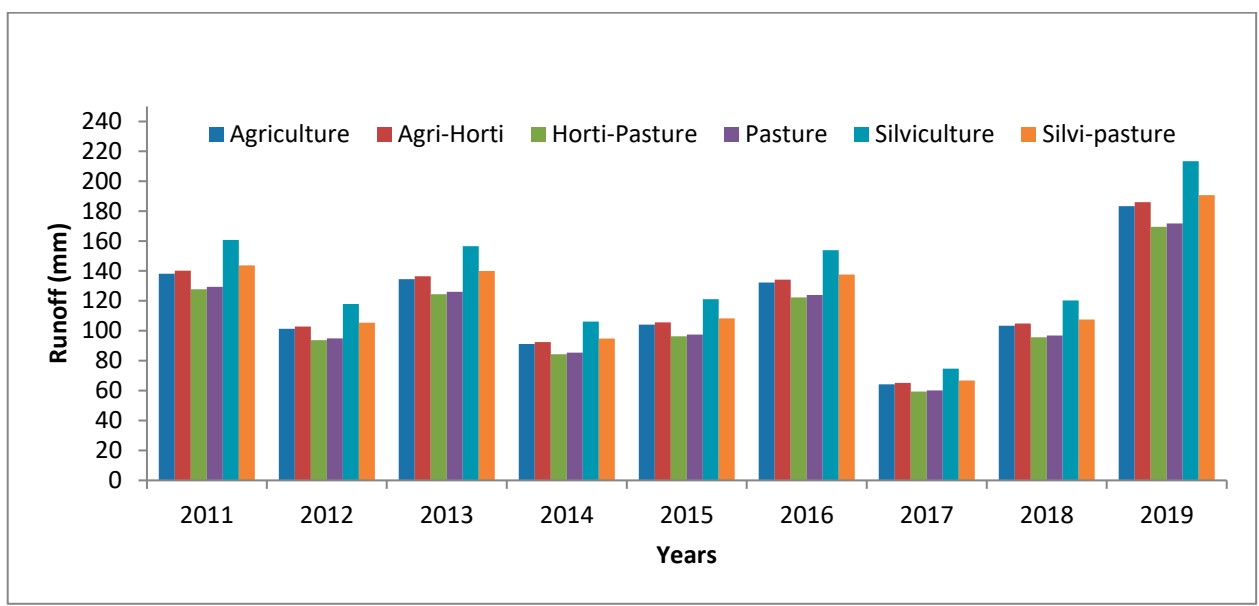

**Figure 8.** Yearly average annual runoff (mm) of each land use systems during 9 years of experiment (2011–2019). $T_1$—Agriculture, *Glycin max.*; $T_2$—Agri-Horti, *Manilkara achras* and *Glycin max*; $T_3$—Horti-Pasture, *Emblica officinalis* and *Cenchrus ciliaris*; $T_4$—Pasture, *Cenchrus ciliaris*; $T_5$—Silviculture, *Accacia nilotica*; $T_6$—Silvipasture, *Accacia nilotica* and *Cenchrus ciliaris*.

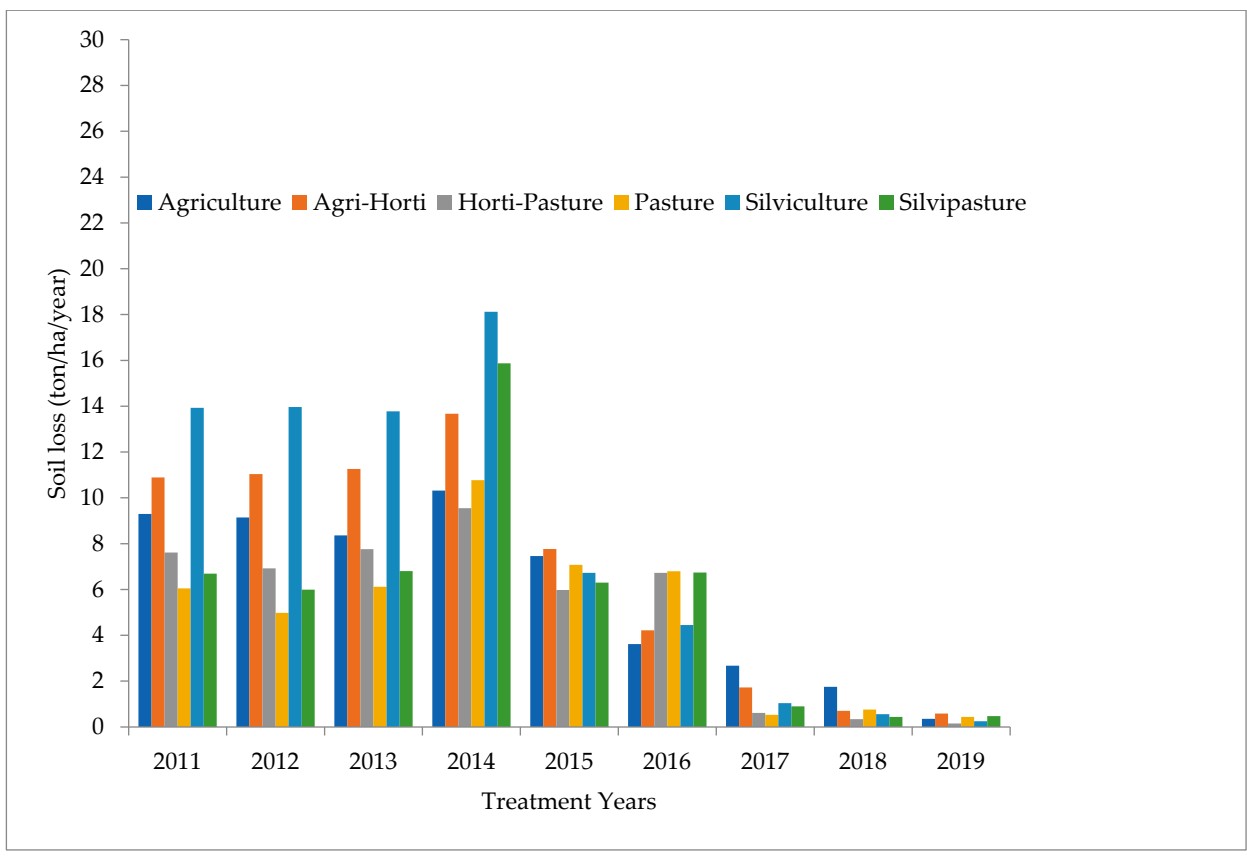

**Figure 9.** Impact of selected land use systems on sediment loss (ton ha$^{-1}$ year$^{-1}$) during the study period. $T_1$—Agriculture, *Glycin max*; $T_2$—Agri-Horti, *Manilkara achras* and *Glycin max*; $T_3$—Horti-Pasture, *Emblica officinalis* and *Cenchrus ciliaris*; $T_4$—Pasture, *Cenchrus ciliaris*; $T_5$—Silviculture, *Accacia nilotica*; $T_6$—Silvipasture, *Accacia nilotica* and *Cenchrus ciliaris*.

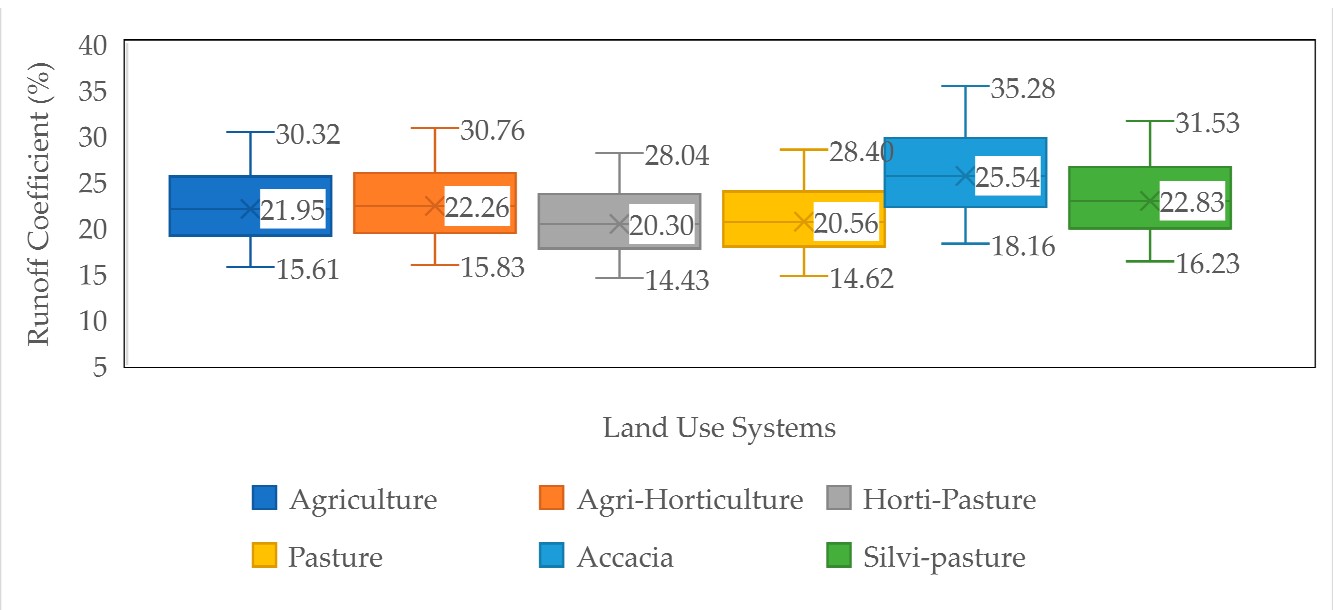

**Figure 10.** Average runoff coefficient (%) of selected land use systems in the marginal lands of Chambal riparian zone of India over the period of nine years of experiment (2011–2019). $T_1$—Agriculture, *Glycin max.*; $T_2$—Agri-Horti, *Manilkara achras* and *Glycin max*; $T_3$—Horti-Pasture, *Emblica officinalis* and *Cenchrus ciliaris*; $T_4$—Pasture, *Cenchrus ciliaris*; $T_5$—Silviculture, *Accacia nilotica*; $T_6$—Silvipasture, *Accacia nilotica* and *Cenchrus ciliaris*.

The layer-wise distribution of available nitrogen and phosphorus showed significant difference at $p = 0.05$, while the available potassium content revealed a non-significant distribution throughout the soil profile (Figure 11). The higher amount of available nitrogen and phosphorus in the top 15 cm soil layer under all selected land use systems can be correlated with the comparatively increased per cent of SOC in the particular soil layer. Our results coincided with Kanthaliya et al. [88] and Xie et al. [89], who reported comparatively increased amounts of nitrogen and phosphorus in the top soil layer along with a higher amount of SOC. The higher amount of and significant variation in available nitrogen and phosphorus in the top soil layer could also be attributed to the continuous addition of organic material and mineralization of added SOM supported by climatic conditions. The available potassium content had a significant bearing on soil depth. The interaction of soil depth and land use did not have a significant impact on available potassium. However, different land uses, especially pasture and plantation, had a significant impact on it at $p = 0.05$, which resulted in a high amount of available potassium throughout the soil profile under the $T_3$ to $T_6$ treatments. It could be attributed to the continuous addition and decomposition of SOM through root biomass, especially fine roots, in those land uses. No significant variation on micronutrients (Fe, Mn, Zn and Cu) was recorded under selected land uses. The runoff revealed a strong positive significant correlation with active carbon and soil loss, and active carbon showed a significant positive correlation with available nitrogen. Similarly to our observations, Garnier et al. [90] reported a linear relationship between dissolved organic carbon and runoff and concluded that the amount of runoff has a direct impact on the lost fraction of active carbon from the total pool of organic carbon in soil. It is a widely accepted and appreciated phenomenon that both carbon and nitrogen are strongly associated and stored with/in soil fractions that are preferably eroded with runoff. Our findings coincide with the findings of Holz and Augustine [91], who reported a significant and positive correlation between runoff and active carbon and nitrogen.

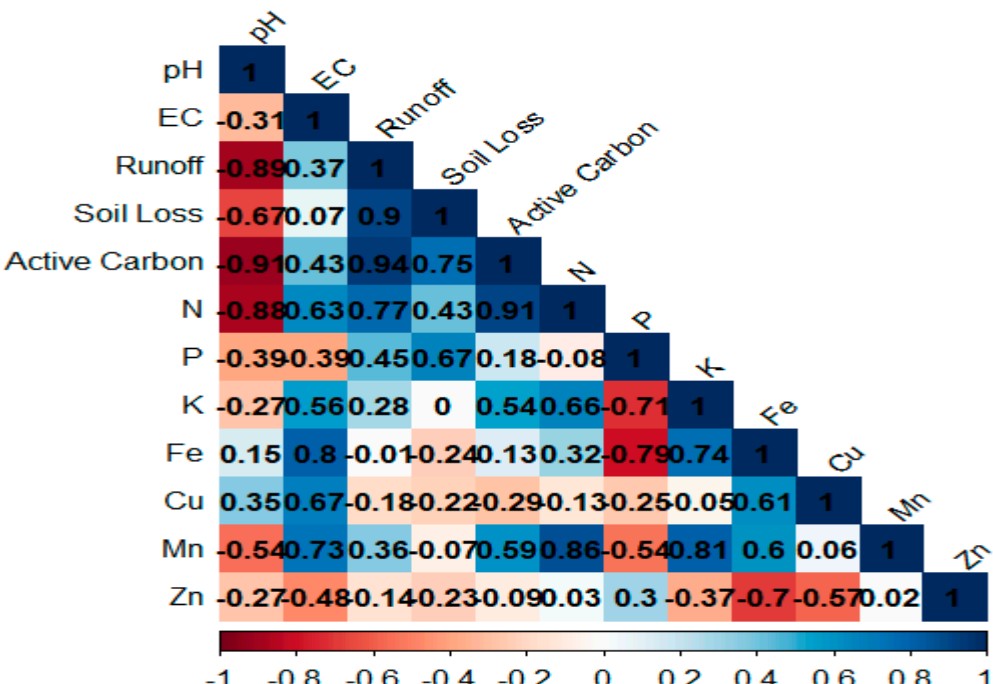

**Figure 11.** Pearson's correlation coefficients between the means of different parameters. Note: Color coding: from white to blue, increasingly greater positive correlation; from white to red, increasing greater negative correlation; white or no color, non-significant correlation (at $p < 0.05$).

## 4. Conclusions

Ecosystem resilience to perturbations that could lead to irreversible land degradation in ecosensitive ravine-like zones in arid and semiarid regions can be addressed up to certain limit by introducing long-term land use systems. Important soil properties (BD, MWD, resistance to penetration and SOC content, etc.) for hydrology and erosional response could be improved under a permanent land use system as opposed to agriculture-like routine land use in ravine-like fragile ecosystems. The Horti-pastoral (*Emblica officinalis + Cenchrus ciliaris*) and T$_4$—Pasture (*Cenchrus ciliaris*)—were identified as the most efficient land use systems for conserving runoff and reducing soil loss. The runoff coefficient under the selected land use systems varies from 20 to 25%, in the following order: Horti-Pastoral (20.30%) < Pasture (20.56%) < Agriculture (21.95%) < Agri-horticulture (22.26%) < Silvi-pasture (22.83%) < Silviculture (25.54%), respectively. The finer fractions of the soil matrix had a considerable role in soil aggregation and soil pH. The *Cenchrus ciliaris* alone and in combination increases the SOC content and MWD, and it decreases bulk density, resistance to penetration, runoff and soil loss in silty clay loam to clay loam soil under semi-arid climatic conditions. We recommended that *Cenchrus ciliaris* can be used alone or in combination with plants for protection of natural resources from water-induced soil erosion, runoff conservation and maximization of precipitation water use for productive purposes. It was found that SOC had a definite role in soil aggregation, but to find out its role in aggregate stability requires further investigation. Land conversion for plantation might lead to stable aggregate buildup compared to tilled fields, but it also augments soil compaction. We hypothesize that it might be due to the formation of a massive or stratified soil matrix in place of an angular and blocky soil structure. However, this fact is not covered under our study, and it needs further study to confirm and justify the impact of converted land use on soil structure patterns. The major environmental services which may be generated by introducing different land use systems in any set-aside degraded ecosystems, such as ravine, mine spoil areas, etc., are depicted in Figure 12.

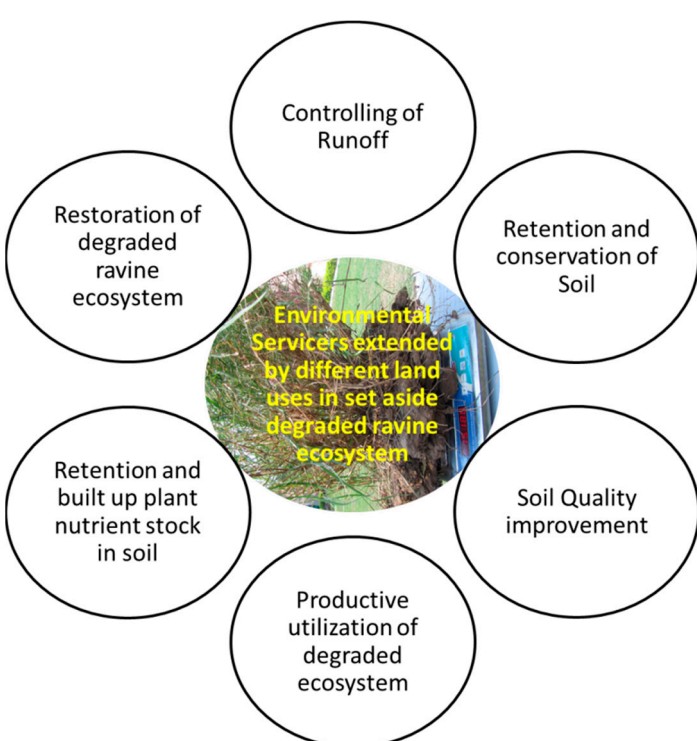

**Figure 12.** Environmental services generated by introducing different land use systems in degraded ecosystems.

**Author Contributions:** Conceptualization, G.L.M. and B.K.S.; methodology, G.L.M. and S.A.; formal analysis, H.R.M. and A.K.; investigation, G.L.M., H.R.M., R.K.S. and B.K.S.; resources, A.K. and S.A.; data curation, G.L.M., B.L.M. and B.K.S.; writing—original draft preparation, G.L.M.; writing—review and editing, G.K.S., R.B.M., B.L.M., K.K. and R.S.M.; visualization, R.S.M., K.K. and R.B.M. All authors have read and agreed to the published version of the manuscript.

**Funding:** This research was funded by the ICAR-Indian Institute of Soil and Water Conservation, Deharadun, Uttrakhand (India). The project code/funding number for this project was NRMACSWCRTI CIL201001000046.

**Institutional Review Board Statement:** Not applicable.

**Data Availability Statement:** The data are available from the corresponding author upon request.

**Acknowledgments:** The authors are thankful to the present and former Director, ICAR-Indian Institute of Soil and Water Conservation, Dehradun, India for providing financial and technical support during project work.

**Conflicts of Interest:** All authors declare that they have no conflict of interest.

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
