# Peer review of "Quantification of Impact of Land Use Systems on Runoff and Soil Loss from Ravine Ecosystem of Western India"

_agriculture, doi:10.3390/agriculture13040773_

Round 1

Reviewer 1 Report

Review on MS agriculture-2203403 submitted to Agriculture

Quantification of Impact of Land Use Systems on Runoff and Soil Loss from Ravine Ecosystem of Western India 

by Gopal Lal Meena et al., 2023

The submitted MS is focused on water induced erosion and present a extensive data from long-term (2011-2019) experiment. Hovewer, MS has very poor writing (redundant sentences, and conversely missed references and detailes, especially in methods section and in footnotes of figures; the text is not divided into paragraphs, where required; many careless mistakes). Thus, it will be possible to say about the value of the material when the authors have revised it significantly, primarily in terms of conciseness and logical narration. Probably, not the entire body of data should be presented in one file, but to divide the material into several subtasks, which you have set in the whole study.

General comments 

Use paragraph breaks and watch out for linkages between sentences. now there is no logic in many places, and the text is not at all clear.

Your plots are overloaded with information, they need to be made simpler.

Figures

Fig. 2 – it is not clear, what is the active SOC (not described in methods) + very hard to read in such form (divide to separate subplots) (the same for fig. 7)

Fig. 3 – dimensions must be written near the axis name + it is not described in methods how you measure it

Fig. 4 -- it is necessary to provide the significance of coefficients and other statistics in supplementary materials

Fig. 6 – not clear, what the gray line means

Tables

BD dimension is not correct

Some specific remarks

L37      an oddly structured sentence, rephrase it. whose demands?

L43-45 missed references

L44      not clear, which type of degradation you describe. The previous sentence is about water erosion, the sentens before and past – about all types of erosion.

L48      new paragraph

L56-57 .”.no obvious relation to climate” ïƒ  missed references

L60      new paragraph

L73-74 examples on “physical and mechanical structures”—not clear what you mean + add references

L82-83 refrace or delete, this sentence is superfluous for the introduction part.

L84-85 missed references

L90-91 reference should me mentioned after words “C. ciliaris grass”

L111    BD – transcribe

L112-115          delete, you will describe it in following sections

L127    «71» incorrect link order

L118-129          check the references, is not clear, where exactly information could be found. Only in ref. 71?

L126    South-Eastern Rajasthan

L139    classify soil by any soil classification (e.g. WRB)

L145-153 add the figure with scheme of the experiment

L159-164 is it the common method of calibration? Than delete extensive details. And add reference in any case.

L190-192 delete up to the words “a stratified purposive soil sampling was”, redundant sentence for the methods; also it is not clear what does “stratified purposive” means, clarify

L199    “using solubridge”— using what?

L200    Walkey and Black. These are the names of the scientists who proposed the method.

L220    redundant sentence

...

Reviewer 2 Report

Comments are provided in the file

Reviewer 3 Report

See attached file

Round 2

Reviewer 2 Report

 Accept in present form

Author Response

Response to the reviewers’ comments: Please find here given response to your constructive comments as:

S. No.

Referee comment

Reply

1         

L. 56. The number of citation should be in order of mention/increase. Instead [73] > [9].

Agreed. Thank you very much for your constructive suggestion. All citation numbers are arranged in increasing order throughout the whole manuscript and references are also cross referenced with bibliography section.

2         

L. 67. Please give a full name of “TGA” abbreviation.

Agreed and addressed.

3         

L. 131. “Soils of the region belong to the ustic soil moisture and hyperthermic temperature regime.” This sentence was already mentioned in the text (L. 128).

Agreed and addressed. Repeated sentence is deleted.

4         

Figure 2. Please improve the quality of picture.

Agreed and addressed accordingly. Improved figure is inserted in manuscript. Blurred photos are also replaced.

5         

L. 181. Full name of instrument “MSD”?

Agreed and addressed accordingly.

6         

L. 186. How you measured soil loss? By calculating of runoff volume and its turbidity?

Soil loss was measured from collected runoff samples which were later transferred into beaker and evaporated at 105° C to assess the event wise soil loss. We can call it weight/volume basis or runoff volume and its turbidity based assessment.

7         

L. 220. “company (Hardware version-6.00, software version-6.03 and penetroviewer version-6.08).” this part of sentence could be omitted.

Agreed and addressed accordingly.

8         

Table 1. g cm3 > g•cm-3. Also in notes instead “#, $, @”, etc you could use *, **, ***, etc.

Agreed and addressed accordingly.

9         

Figure 3. Doil > Soil.

Agreed and addressed accordingly.

10      

Figure 4. Mpa > MPa. In soil depth please delete “-“ (minis), also here please round to integer the values (eg. 2.0 > 2).

Agreed and addressed accordingly. Improved figure is inserted in manuscript.

11      

Figure 5. The correlation between pH and fine particles did for which soil layer?

Correlation between pH and fine particles was assessed on profile basis (0-60 cm soil depth) for average values of fine particles. It was not on horizon wise. This is well established fact that soil texture is permanent property of soil and not easily changed over such a small period of time. So, we consider whole profile as a unit. It is also mentioned in result and discussion section.

12      

L. 415. reason > season

Agreed and addressed accordingly.

13      

L. 424. 2011-14 > 2011-2014

Agreed and addressed accordingly.

14      

Figure 7 and 8. What do you mean under runoff % (how you measured it?), usually runoff presented in “mm”.

Figure 7 – you use name of treatments T1, T2… better if the other figures will presented in same way.

Agreed. We also measured runoff in mm then converted it into per cent (%). It is ratio of the event runoff (mm) to the event rainfall (mm).

Regarding treatments T1, T2… in figure 7, it is humbly submitted that during first review of the manuscript, reviewers suggested that treatments name should be mentioned in all figures rather than T1, T2… So, I followed the same suggestion for all figures except figure 7 which was left by mistake. Sorry for that. I discussed with all team members and they also suggest me to mention treatment names rather than T1, T2…. So, I replaced figure 7 with improvement. It is requested to you that if you agree with treatments name in figure then please accept it or if you desire then we replace all figures.  
